# Scoring the Sit-to-Stand Performance of Parkinson’s Patients with a Single Wearable Sensor

**DOI:** 10.3390/s22218340

**Published:** 2022-10-30

**Authors:** Frédéric Marin, Elke Warmerdam, Zoé Marin, Khalil Ben Mansour, Walter Maetzler, Clint Hansen

**Affiliations:** 1Laboratoire de BioMécanique et BioIngénierie (UMR CNRS 7338), Centre of Excellence for Human and Animal Movement Biomechanics (CoEMoB), Université de Technologie de Compiègne (UTC), Alliance Sorbonne Université, 60200 Compiègne, France; 2Department of Neurology, Kiel University, 24105 Kiel, Germany; 3Faculty of Computer Science or Communication Systems, Ecole Polytechnique Fédérale de Lausanne (EPFL), CH-1015 Lausanne, Switzerland

**Keywords:** Parkinson’s disease, sit-to-stand, IMU, motion analysis, movement biomechanics

## Abstract

Monitoring disease progression in Parkinson’s disease is challenging. Postural transfers by sit-to-stand motions are adapted to trace the motor performance of subjects. Wearable sensors such as inertial measurement units allow for monitoring motion performance. We propose quantifying the sit-to-stand performance based on two scores compiling kinematics, dynamics, and energy-related variables. Three groups participated in this research: asymptomatic young participants (n = 33), senior asymptomatic participants (n = 17), and Parkinson’s patients (n = 20). An unsupervised classification was performed of the two scores to differentiate the three populations. We found a sensitivity of 0.4 and a specificity of 0.96 to distinguish Parkinson’s patients from asymptomatic subjects. In addition, seven Parkinson’s patients performed the sit-to-stand task “ON” and “OFF” medication, and we noted the scores improved with the patients’ medication states (MDS-UPDRS III scores). Our investigation revealed that Parkinson’s patients demonstrate a wide spectrum of mobility variations, and while one inertial measurement unit can quantify the sit-to-stand performance, differentiating between PD patients and healthy adults and distinguishing between “ON” and “OFF” periods in PD patients is still challenging.

## 1. Introduction

Parkinson’s disease (PD) is the second most common neurodegenerative disease [1], and in light of an aging global population, the expected number of PD patients will double by 2030 [2]. PD is a progressive neurodegenerative disorder [3] characterized by both motor and nonmotor symptoms [4]. Cardinal motor symptoms are rest tremors, bradykinesia, and rigidity [3] at various levels of intensity and frequency. The disease progression is often accompanied by a loss of postural reflexes, freezing of gait, and a stooped posture [3]. Nonmotor symptoms include cognitive and psychological deficits [5]. Altogether, these symptoms affect the quality of life and reduce the autonomy of the patient [5]. Since PD is highly patient-specific, symptom progression is very individual [6].

The diagnosis of PD is based on the cardinal criteria, i.e., presence of bradykinesia and at least one of either tremor or rigidity [7]. The Hoehn and Yahr stage (HY) [8] and the revised version of the MDS-Unified Parkinson’s Disease Rating Scale (MDS-UPDRS) [8] are scales to evaluate the presence and intensity of symptoms. Levodopa (L-Dopa), a dopamine precursor, is still the most effective drug to treat PD. While the drug is working, PD patients experience “ON” periods leading to improved mobility [9], but when the drug does not work optimally, patients enter an “OFF” period, and the motor and nonmotor symptoms increase in severity. PD management requires constant observation of the symptoms’ progression [10], and when patients exhibit a waning response to the medication, drug doses have to be readjusted [11].

One motion that allows for the quantification of mobility is a sit-to-stand (STS) movement. STS movements are (1) frequently performed throughout the day (approx. 60 times (±22) in healthy adults [12]), (2) exhibit a transfer of the upper body, and (3) require a high degree of coordination [13]. Neurological disease progression can be measured with STS [14] and related to subtle alterations of the musculoskeletal system [15] and the risk of falls [16] in PD patients. Consequently, the quality of STS movements is a good indicator of independence [17], leg muscle strength [18], and postural control [19]. A survey of 101 PD patients reported that 81% have trouble performing STS [5], and STS performance differs according to PD subgroups [20] and medication states (“ON” and “OFF” periods) [21].

Quantifying STS performance and relating specific metrics to the aforementioned independence, muscle strength, and risk of falls remains challenging [22]. Three-dimensional motion capture systems [13] and force plates [23] can quantify STS performance in the research environment. However, for clinical routines and follow-up examinations at home, wearable sensors/inertial measurement units (IMUs) [23,24] are a valid alternative [25]. They are an ideal tool to capture postural transitions [26] and STS [22,27,28] both in clinical practice as well as at home.

Recently, LePetit et al. [29] proposed a method to quantify STS performance based on two scores. The *A*-score and the *f*-score are combinations of kinematics, dynamics, and energy-related variables from one single IMU. These scores resulted in an interactive process to determine which set of kinematic, dynamic, or energy-related variables best differentiated the studied population. An analysis of the area under the curve (AUC) of the receiver operating characteristic (ROC) was computed for each combination of the set of variables. The scores allowed for the differentiation between the frail and the healthy amongst old individuals and between the old and the young amongst healthy individuals, which maximized the AUC.

As the differentiation between frail and healthy seniors was successful, this study aimed to apply the scores to PD patients and healthy adults and relate them to the clinical scores and medication state. We hypothesized that these scores could differentiate between PD patients and healthy adults and also distinguish between “ON” and “OFF” periods in PD patients.

## 2. Materials and Methods

### 2.1. Participants

Twenty PD patients and fifty asymptomatic participants enrolled in this study (Table 1).

Asymptomatic participants were recruited via flyers that were placed in public facilities and divided into two groups: young adults (18–60 years) and senior adults (>60 years). The PD patients were recruited from either the outpatient clinic or the neurology ward of the University Hospital Schleswig-Holstein, Campus Kiel, Germany. The inclusion criterion for the PD group was a Parkinson’s diagnosis according to the UK Brain Bank criteria [30]. Subjects were excluded who used a walking aid and had a Montreal Cognitive Assessment (MoCA) score below 15. In addition, for asymptomatic groups, subjects were excluded if they had a movement disorder that was not age-related or if they reported any pain. For the PD patients, subjects were excluded who had a movement disorder besides their primary diagnosis. Ten patients were measured during medication “OFF” (PD_off_), seventeen during medication “ON” (PD_on_), and seven were measured during both medication “ON” and medication “OFF” periods. In addition, two groups of healthy participants were included: 33 asymptomatic young adults (A_Y_; 18–60 years) and 17 asymptomatic senior adults (A_S_; 60+ years).

For all participants, a trained clinician assessed the motor section of the MDS-UPDRS (part III). For the PD participants, the MDS-UPDRS III was assessed in each medication state for which the participant was measured.

### 2.2. Protocol

An IMU (Noraxon USA Inc., Scottsdale Arizona, AZ, USA) including a 3D accelerometer and a 3D gyroscope was fixed on the thorax by elastic straps worn around the upper part of the torso (Figure 1).

Technical calibration was performed to register the local reference frame (S) of the IMU with the anatomical axes of the torso (T), i.e., proximal–distal (*PrD*), medio–lateral (*ML*) and antero–posterior (*AP*) axes [22]. Each participant sat at a standard seat height with a knee angle of around 90° and both feet firmly on the ground. At the beginning of the session, the participants sat quietly for around 10 s. Then, participants were asked to perform the five STS tests at their preferred pace without using their arms. During five chair-rises, the IMU recorded accelerations and angular velocities in the local reference frame of the sensor with a sampling rate of 200 Hz. At the end of the session, the participant recovered by sitting quietly for a few seconds, and the data collection was ended. The rationale of sensor placement and protocol specification are described in Warmerdam et al. (2021) [31].

The data collected were part of a larger project [31] approved by the ethical committee of the Medical Faculty of Kiel University (D438/18) and in accordance with the principles of the Declaration of Helsinki. All participants received written and oral information about the measurements. The participants provided written informed consent before the start of the measurements. The study was registered in the German Clinical Trials Register (DRKS00022998). Some PD patients who consented to assessments during ON and OFF dopaminergic medication states were measured in both conditions. This took extra time, as both the assessors as well as the participants had to wait for the dopaminergic medication to take effect and had to perform the whole protocol twice.

### 2.3. Postprocessing

To quantify the STS performance of the first movement, the *a*-vector of the AgingScore and *f*-vector of the FrailtyScore were computed [29] as follows.

First, using a fusion algorithm [32] at each time t, the linear acceleration and angular velocity were computed in the global reference frame (G), i.e., down–up (DU), backward–forward (BF), and right–left (RL)), respectively:$$a_{G}\left(t\right)=\left[\begin{array}{ccc} a_{DU}\left(t\right) & a_{BF}\left(t\right) & a_{RL}\left(t\right) \end{array}\right]\;\;\mathrm{and}\;\;\omega_{G}\left(t\right)=\left[\begin{array}{ccc} \omega_{DU}\left(t\right) & \omega_{BF}\left(t\right) & \omega_{RL}\left(t\right) \end{array}\right]$$

Then, linear accelerations and angular velocities were computed in the torso reference frame [29]:
$$a_{T}\left(t\right)=\left[\begin{array}{ccc} a_{PrD}\left(t\right) & a_{ML}\left(t\right) & a_{AP}\left(t\right) \end{array}\right]\;\mathrm{and}\;\omega_{T}\left(t\right)=\left[\begin{array}{ccc} \omega_{PrD}\left(t\right) & \omega_{ML}\left(t\right) & \omega_{AP}\left(t\right) \end{array}\right]$$

In addition, ${\mathrm{Vg}}_{T},$ the velocity of the center of gravity of the torso, and the kinetic energy (EK) of the torso, were computed [22].

Once the timing of the beginning $t_{b}$ and the end $t_{bf}$ of STS were determined [22] for each participant s $\in \left\{{\mathrm{PD}}_{\mathrm{off}},{\;\mathrm{PD}}_{\mathrm{on}},A_{Y},A_{S}\right\}$, we defined the a-vector$\left(s\right)$ as:


$$\begin{array}{c} a-\mathrm{vector}\;(s)\;=\;[\mathrm{maxAcc},\;\mathrm{maxAz},\;\mathrm{maxAxy},\;\mathrm{maxVG},\;\mathrm{maxOmega}],\;\;\mathrm{with}\;\\ \mathrm{maxAcc}=\max_{t\in \left[t_{b},t_{f}\right]}\left(\sqrt{{\left(a_{DU}\left(t\right)\right)}^{2}+{\left(a_{BF}\left(t\right)\right)}^{2}+{\left(a_{RF}\left(t\right)\right)}^{2}}\right),\\ \mathrm{maxAz}=\max_{t\in \left[t_{b},t_{f}\right]}\left(\left|a_{DU}\left(t\right)\right|\right),\\ \mathrm{maxAxy}:=\max_{t\in \left[t_{b},t_{f}\right]}\left(\sqrt{{\left(a_{BF}\left(t\right)\right)}^{2}+{\left(a_{RF}\left(t\right)\right)}^{2}}\right),\\ \mathrm{maxVG}\;=\max_{t\in \left[t_{b},t_{f}\right]}\left(∥{\mathrm{Vg}}_{T}\left(t\right)∥\right),\;\mathrm{and}\;\\ \;\mathrm{maxOmega}\;=\max_{t\in \left[t_{b},t_{f}\right]}\left(\sqrt{{\left(\omega_{DU}\left(t\right)\right)}^{2}+{\left(\omega_{BF}\left(t\right)\right)}^{2}+{\left(\omega_{RF}\left(t\right)\right)}^{2}}\right)\;\;\\ \;\mathrm{and},\;\;f-\mathrm{vector}\;\left(s\right)=\;\left[\mathrm{mVG},\;\mathrm{mEK},\;\mathrm{mAz},\mathrm{TD},\;\mathrm{maxEK},\;\mathrm{mAcc},\;\mathrm{AUCml}\right]\;\;\mathrm{with}\\ \mathrm{mVG}\;=\underset{t\in \left[t_{b},t_{f}\right]}{\mathrm{mean}}\left(∥{\mathrm{Vg}}_{T}\left(t\right)∥\right)\;\;\;\\ \mathrm{mEK}\;=\underset{t\in \left[t_{b},t_{f}\right]}{\mathrm{mean}}\left(\mathrm{Ek}\left(t\right)\right),\;\;\\ \mathrm{mAz}=\underset{t\in \left[t_{b},t_{f}\right]}{\mathrm{mean}}\left(\left|a_{DU}\left(t\right)\right|\right),\\ \;\mathrm{TD}\;=t_{f}-t_{b},\;\;\\ \mathrm{maxEK}=\max_{t\in \left[t_{b},t_{f}\right]}\left(\mathrm{Ek}\left(t\right)\right),\\ \;\mathrm{mAcc}\;=\underset{t\in \left[t_{b},t_{f}\right]}{\mathrm{mean}}\left(\sqrt{{\left(a_{DU}\left(t\right)\right)}^{2}+{\left(a_{BF}\left(t\right)\right)}^{2}+{\left(a_{RF}\left(t\right)\right)}^{2}}\right)\;\;,\mathrm{and}\;\\ \mathrm{AUCml}\;=\int_{t_{b}}^{t_{f}}\left|a_{ML}\left(t\right)\right|dt \end{array}$$


In summary, the *a*-vectors are composed of the maximal norm of the acceleration during the STS (maxAcc), the maximal absolute values of the up–down acceleration (maxAz) and the horizontal plane (maxAxy) of the torso, the maximal value of the velocity of the torso (maxVG), and the maximal value of the norm of the rotational velocity of the torso (maxOmega). The *f*-vector is composed of the mean value of the velocity of the torso (mVG) during the STS, the mean value of the kinetic energy (mEK), the mean value of the absolute value of the up–down acceleration (mAz), the duration of the STS (TD), the maximal value of the kinetic energy (maxEK), the mean value of the norm of the acceleration during the STS (mAcc), and the area under the curve of the absolute value of the medio–lateral acceleration (AUCml).

The parameters were chosen based on their discrimination performance [29]. In short, the authors used the area under the curve (AUC) of a receiver operating characteristic (ROC) curve with the aim to reduce the k-length vector to a scalar-based score. This was done using an iterative principal component analysis (PCA) procedure. The first principal component, PC1, maximizes the variance in one dimension and has the highest potential in terms of classification accuracy. The combination of parameters maximizing the classification accuracy associated with aging and frailty defined the a-score and f-score [29].

### 2.4. Statistical Analysis

A principal component analysis (PCA) with a standardized correlation matrix was conducted with the a-vectors and f-vectors of all participants. The first principal component of the a-vectors for each participant was defined as the a-score [29]. The a-score is a linear combination of the component of the a-vector by the determination of coefficients according to the PCA procedure. In the same way, based on the PCA with all f-vectors, the f-score for each participant was defined as the first principal component [29]. Then, the performance of STS for each participant was analyzed on the a-score vs. f-score plane. K-means clustering was performed to partition all participants in the a-score vs. f-score plane [33]. Three clusters were computed after 30 repetitions of the iterative clustering algorithm to avoid the convergence to a local minimum using the k-means clustering function (kmeans) in MATLAB software (MATLAB R2021b, The MathWorks, Inc., Natick, MA, USA). We assumed that if the a-score vs. f-score plane is representative of the STS performance, then the three groups of participants, i.e., asymptomatic participants (young and senior) and PD, would be separately classified into three clusters. Calculations of the sensitivity and specificity were performed according to the classification of asymptomatic subjects and PD patients.

Linear regression analyses were performed to evaluate the relationship between the UPDRS score and the *a*-score and *f*-score. The linear regression computes the best-fitting straight line to the data points that best characterizes the relationship between a dependent variable *Y*, i.e., *a*-score or *f*-score and the independent variable *X*, i.e., UPDRS score defined by the slope *k* and the intercept *Y*0 [34], as follows:Y=kX+Y0

In addition, a multivariable linear regression model was also computed, where the independent variables were V_1_ and V_2_, which are the *a*-score and the *f*-score, respectively, and the dependent variable was *W*, which is the UPDRS score, with slopes *k*1 and *k*2 and the intercept W0 as follows:$$W=k1V_{1}+k2V_{2}+W0$$

To quantify the relevance of linear regression, we also computed the 95% confident intervals of the slope and the intercept, the coefficient of determination R^2^, which quantifies the proportion of the variability in the dependent variable explained by the independent variable, and the *p*-values of the *F*-test, which estimated the significance level of the linear regression (traditionally, the linear regression is statistically significant if *p* < 0.05) [35].

In addition, for the seven PD patients that were measured in both “OFF” and “ON”, improvement or worsening of the STS performance was quantified on the gradient $\pm \Delta_{\mathrm{af}}$, with *a*-scores and *f*-scores obtained by the same subject at the “OFF” and “ON” stage. Let a subject have scores [$a{-\mathrm{score}}_{OFF}$, $f{-\mathrm{score}}_{OFF}$] at the “OFF” stage and [$a{-\mathrm{score}}_{ON}$, $f{-\mathrm{score}}_{ON}$] at the “ON” stage. We could deduce:$$\pm \Delta_{\mathrm{af}}\;=\;sign\left(f{-\mathrm{score}}_{ON}-f{-\mathrm{score}}_{OFF}\right)\times \sqrt{{\left(a{-\mathrm{score}}_{ON}-a{-\mathrm{score}}_{OFF}\right)}^{2}+{\left(f{-\mathrm{score}}_{ON}-f{-\mathrm{score}}_{OFF}\right)}^{2}}$$

A positive gradient ($+\Delta_{\mathrm{af}}$) could be associated with an improvement in the STS performance of the subject at stage “ON”, and a negative gradient ($-\Delta_{\mathrm{af}}$) as a worsening in STS. The assumption was that ($+\Delta_{\mathrm{af}}$) is associated with a decrease in the UPDRS score ($-\Delta_{\mathrm{UPDRS}}$) between the “ON” and “OFF” states of the patients.

## 3. Results

Based on the PCA of the a-vectors and f-vectors of all participants, the a-score and the f-score were the first principal components that maximized the percent of variability explained, with 67.6% and 64.1%, respectively. Coefficients of the a-score (Table 2) and the f-score (Table 3), which are values of the linear composition with the components of the a-vectors and f-vectors according to the PCA procedure, demonstrated a homogeneity in the weight of each component of the a-vectors and f-vectors, with an exception for the parameter AUCml.

In the a-score vs. f-score plane (Figure 2), three clusters were stratified according to the a- and f-scores. We identified an “upper” and a “lower” cluster based on the highest and lowest values of the a- and f-scores. In between both clusters, an “intermediate” cluster was defined.

The “upper” cluster mostly contains $A_{Y}$ participants but also five $A_{S}$ and six PD patients (Table 4). The “intermediate” cluster includes the majority of participants in the $A_{S}$ group. The “lower” cluster contains mainly PD participants. According to this cluster repartition, we obtained a specificity of 0.96 and a sensitivity of 0.4 of the classification of the PD subjects (ON and OFF) in the “lower” cluster, relative to all asymptomatic subjects.

Results of the linear regression analyses of the a-score vs. UPDRS score and f-score vs. UPDRS score are summarized in Figure 3 and Table 5.

Linear regressions of the *a*-score vs. UPDRS score and *f*-score vs. UPDRS score were statistically significant (*p* < 0.05). However, we noted a low coefficient of determination (R^2^). Only 28% and 24% of the variability in the *a*-score and *f*-score, respectively, were explained by their relationship with the UPDRS score.

For the multivariable linear regression model, we found the slope k1 of −2.740 (with 95% CI from −5.082 to −0.398) for the independent variable V_1_, associated with the *a*-score, the slope k2 of −1.210 (with 95% CI from −3.243 to 0.823) for the independent variable V_2_, associated with the *f*-score, and intercept W0 of 9.961 (with 95% CI from 7.370 to 12.551). For this model. we obtained a *p*-value of 2.676 × 10^−6^ and an R^2^ value of 0.29.

When specifically looking at the seven PD patients performing the STS in “ON” and “OFF” medication states (Table 6), five showed an improvement in the performance of the STS in the a-score vs. f-score plane, showing a positive gradient ($+\Delta_{af}$). For two participants, we noted a regression shown by a negative gradient ($-\Delta_{af}$) (Table 6).

In addition, the variation of the MDS-UPDRS III score correlated with the gradient $\Delta_{af}$ (Figure 4). In two cases, we had a status quo of the MDS-UPRDS III, which was associated with either a very slight positive gradient or a negative one.

## 4. Discussion

The present study investigated STS performance measured by a single wearable sensor and its association with clinical scores and medication states. In comparison with young and senior asymptomatic participants, PD patients presented lower quantitative scores. We found a sensitivity of 0.4 and a specificity of 0.96 in distinguishing Parkinson’s patients from asymptomatic participants.

STS movements are good indicators of the quality of life and musculoskeletal functions, and they are easy to perform both in clinical practice and at home [36]. Traditionally, only the duration of the five chair-rise test is used, which is insufficient for a complete clinical performance evaluation [37]. STS movements are complex, requiring balance and strength [38]. Several factors are known to decrease STS performance, e.g., age [36,38,39], back pain [40], obesity [41], and frailty [29]. We, therefore, selected a multidimensional approach, which allowed for the computation of an *a*-score and *f*- score [29]. These scores are a linear combination of kinematic, dynamic, and energetic variables extracted from the IMU raw sensor data [29], which can document modification of the STS strategy, e.g., limitation of torso flexion in the case of high-BMI subjects [42] or augmentation of the duration of STS in older people [36]. Our results showed that young asymptomatic participants had the highest scores (i.e., the “upper” cluster), and senior asymptomatic participants mostly had intermediate scores (i.e., the “intermediate” cluster). However, our results did not meet expectations, as LePetit et al. [29] found a sensitivity and specificity of 0.9 between the senior and frail population, but we found a sensitivity of only 0.4. One explanation for the low sensitivity could be that the *a*-score and *f*-score were initially designed for a population of senior frail subjects, leading us to suggest the development of specific score for PD subjects.

In fact, collectively, PD participants had lower or equivalent *a*-scores and *f*-scores in comparison with the asymptomatic participants. This observation supports the fact that PD is a disease with very individual characteristics and a large spectrum of symptoms and severity levels [6]. We also observed a relationship (linear regression and multivariable linear regression) between *a*-scores and *f*-scores and the MDS-UPDRS III score, which is in accordance with the component of the MDS-UPDRS III score focused on motoric examination [8]. However, in addition to that, we noted a low level of the coefficient of determination of linear regression, which could be explained by the fact that the MDS-UPDRS III score included several additional components of motoric examination, including facial expression, rigidity, hand movement, and leg agility [8]. In addition, only four out of seven PD patients showed an improvement in STS performance in their “ON” phase. This could be partially explained by the fact that the L-Dopa response ranged from improving to worsening of the mobility of PD patients [43] without taking into account L-Dopa-induced side effects [43]. In general, 50% of all PD patients experience diphasic dyskinesia and dystonia due to L-Dopa administration [44], which was not observed in this study when looking at the correlation between the MDS-UPDRS III score and the *a*-scores and *f*-scores. However, the correlation could be flawed, as the MDS-UPDRS III scale is not adapted to discriminate between the wide spectrum of symptoms [45,46].

The current study has potential limitations. Only on the first of the five consecutive STS was used in the analysis to limit the effect of fatigue [47] and rhythmic stimulation [48]. Furthermore, the generalization of our findings is limited by the sample size of twenty PD subjects and fifty asymptomatic subjects and by the fact that only seven participants were measured during medication “ON” and “OFF” periods.” However, our results do demonstrate a trend and could thus serve as a pilot and hypothesis-generating study, which could be confirmed in larger follow-up studies.

## 5. Conclusions

The present study investigated the ability to quantify the STS performance of PD patients using a single IMU as a wearable sensor. A multidimensional approach was used to quantify the performance of the STS motion based on two scores that combine kinematic, dynamic, and energy-related variables. The classification results did not meet our expectations. Both scores could only roughly differentiate PD patients from asymptomatic subjects. Further studies could focus on concatenating multiple scores derived not only from STS but also from other tests, e.g., the timed get-up-and-go (TUG) unipodal test. New tools based on machine learning models seem promising but still require a large harmonized database [49]. However, for the seven PD patients who were measured in medication “ON” and “OFF” periods, the performance improvement was negatively correlated with the MDS-UPDRS III score. This is encouraging. The use of a single wearable sensor is convenient for the participant and has the potential to be included easily in routine clinical assessment. Hence, the combination of a single wearable sensor with new PD-specific scores could be a good indicator of medication states and a good measure/biomarker of treatment efficacy as defined by the FDA (FDA-NIH Biomarker Working Group 2016).

## Figures and Tables

**Figure 1 sensors-22-08340-f001:**
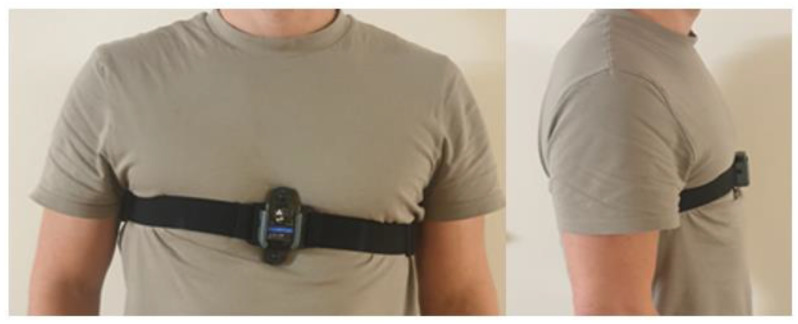
Placement of the inertial measurement unit on the body as described in Warmerdam et al., (2021) [31].

**Figure 2 sensors-22-08340-f002:**
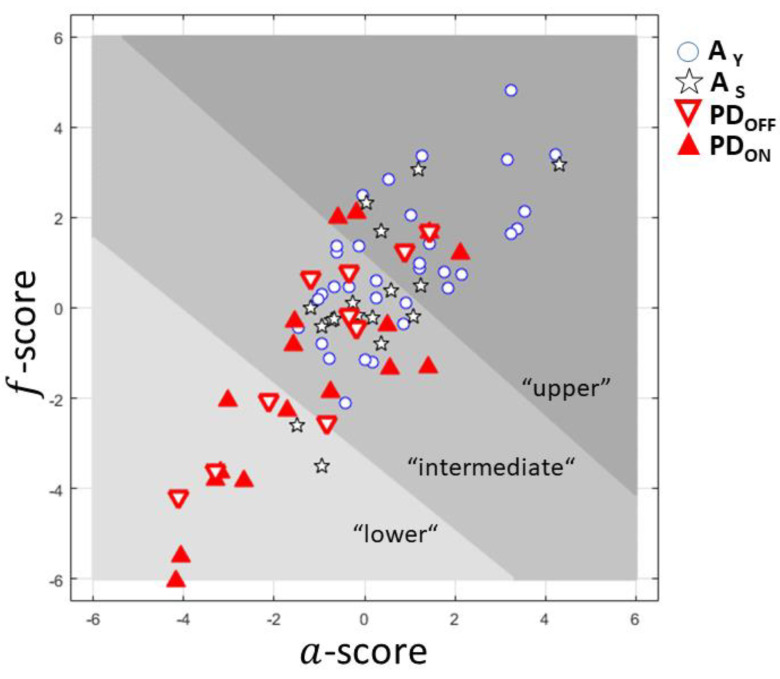
The *a*-score vs. *f*-score plane for asymptomatic subjects (A_Y_ and A_S_) and PD subjects (PD_off_ and PD_on_).

**Figure 3 sensors-22-08340-f003:**
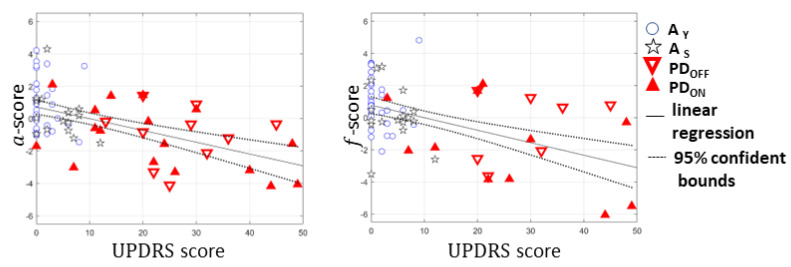
Linear regression with 95% confident bounds of *a*-score vs. UPDRS score and *f*-score vs. UPDRS score.

**Figure 4 sensors-22-08340-f004:**
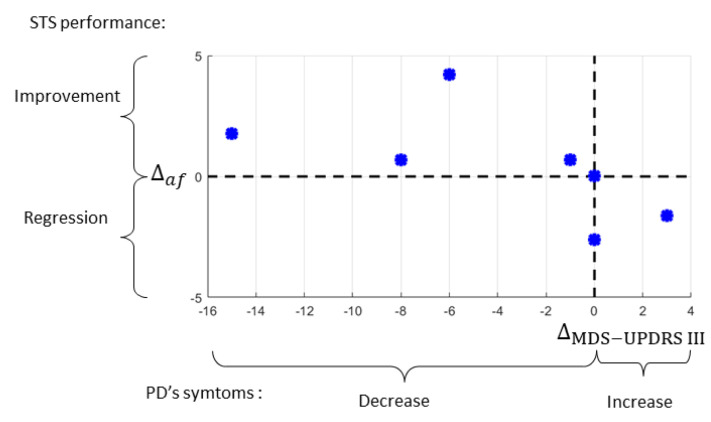
Variation of MDS-UPDRS III score vs. the gradient $\Delta_{af}$ plane.

**Table 1 sensors-22-08340-t001:** Details of the groups involved in the study. The value between parentheses represents the standard deviation. PD_off_ = PD patients in “OFF” medication state; PD_on_ = PD patients in “ON” medication state; A_Y_ = asymptomatic young participants; A_S_ = asymptomatic senior participants.

Group	n♀	n♂	Age Years	BMI kg/cm^−2^	MDS-UPDRS III	Disease Duration Years
${\mathrm{PD}}_{\mathrm{off}}$	3	7	59.5 (9.5)	27.6 (2.6)	27 (9)	9.3 (6.3)
${\mathrm{PD}}_{\mathrm{on}}$	8	9	62.2 (10.7)	26.9 (3.2)	22 (15)	7.9 (4.9)
$A_{Y}$	13	20	28.7 (7.6)	22.3 (2.9)	1 (2)	-
$A_{S}$	5	12	72.0 (8.1)	25.5 (3.8)	4 (4)	-

**Table 2 sensors-22-08340-t002:** Coefficients of the *a*-score in relation to the component of the *a*-vector. Details of the parameters used are in Section 2.3.

Component of the a	maxAcc	maxAz	maxAxy	maxVG	maxOmega
Coefficients of a-score	0.5142	0.4634	0.4518	0.4145	0.3807

**Table 3 sensors-22-08340-t003:** Coefficients of the *f*-score in relation to the component of the *f*-vector. Details of the parameters used are in Section 2.3.

Component of the f	mVG,	mEK,	mAz,	TD,	maxEK,	mAcc,	AUCml
Coefficients of f-score	0.4314	0.4128	0.4354	−0.3407	0.3916	0.4199	−0.0904

**Table 4 sensors-22-08340-t004:** Confusion matrix of the repartition of subjects into clusters.

Cluster	n $A_{Y}$	n $\;A_{S}$	n $\;PD_{off}$	n $\;PD_{on}$
upper	17	5	2	4
intermediate	16	10	4	6
lower	0	2	4	7
n total	33	17	10	17

**Table 5 sensors-22-08340-t005:** Results of the linear regression analyses.

Dependent Variable	Independent Variable	Slope	95% CI Slope	Intercept	95% CI Intercept	R^2^	*p*-Value
a-score	UPDRS score	−0.073	−0.099 to −0.046	0.724	0.278 to 1.169	0.28	7.67 × 10^−7^
f-score	UPDRS score	−0.077	−0.109 to −0.046	0.774	0.247 to 1.300	0.24	5.78 × 10^−6^

**Table 6 sensors-22-08340-t006:** Focus on seven PD patients performing the STS in “ON” and “OFF” medication states.

Participant ID	Sex	Age	BMI Years	MDS-UPDRS III OFF	MDS-UPDRS III ON	Disease Duration Years	$\Delta_{af}$
pp022	♂	52	26.6	20	20	9	0.02
pp045	♂	59	28.2	45	48	4	−1.62
pp082	♂	48	24.2	36	21	7	1.77
pp102	♀	63	20.6	30	30	7	−2.62
pp104	♀	77	28.6	20	12	2	0.68
pp038	♂	64	27.7	13	11	10	0.70
pp046	♂	61	29.8	25	24	18	4.22

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
