# Peer review of "Scoring the Sit-to-Stand Performance of Parkinson’s Patients with a Single Wearable Sensor"

_sensors, 2022, doi:10.3390/s22218340_

Round 1

Reviewer 1 Report

The authors proposed using a single wearable inertia sensor to assess the sit-to-stand performance of people with Parkinson's disease using the frailty and aging scores. The differences in those two scores between PD patients and healthy controls were compared to test the method.

The paper is well written, with sufficient details for each session.

Minor concerns:

1. Add inclusion/exclusion criteria for the two healthy control groups is suggested.

2. Were there two data collection sessions? How were the seven PD patients assessed with both medication ON and OFF?

3. The sensor position was referred to in another paper. It would be great to briefly state the sensor position protocol instead of asking the readers to read another paper to get critical information.

Reviewer 2 Report

Thanks for the opportunity to review the manuscript titled, " Scoring the sit-to-stand performance of Parkinson's patients with a single wearable sensor". The manuscript aims to evaluate the validity of using 1 IMU to detect the sit-to stand performance in people with Parkinson's disease.

Results of the study showed that 2 indexes obtained by using PCA on IMU data demonstrated a high specificity but low sensitivity to differentiate between people with PD and healthy subjects.

The study may support using a single IMU to detect sit-to-stand performance in people with PD or other disease groups with mobility deficits.

1.       My major concern is that the validity of the classification has not been well-evaluated. I would suggest the authors compare the UPDRS score for those PD patients who were classified in ‘lower function’ cluster to those who were classified in the ‘upper and intermediate’ cluster and discuss the findings.

Other comments:

2.       Line 64 please provide more information regarding the discriminated validity reported by LePetit et.al.

3.       Try to justify the sample size

4.       I find it difficult to interpret all the abbreviations from Line 128 to Line 142, The authors can provide a remark on the list of abbreviations after line 142 for easy reference.

5.       Line 170 -171: Can the authors provide a figure to illustrate how to calculate the gradient (Δaf) for quantifying the improvement of STS ?

6.       Line 179: the statement ‘demonstrated a homogeneity of the 179 weight of each component of the ?-vectors and ?-vectors’ may not be correct as the factor loading of AUCml is very low.

7.       Typo: Line 199: ‘Sensibility’

8.       In the discussion (Line 238) the author can compare their results with that reported by LePetit et.al.

9.       In view of the high specificity but low sensitivity, can the authors propose some solution to improve the sensitivity?

Reviewer 3 Report

The ArticleScoring the sit-to-stand performance of Parkinson's patients with a single wearable sensor interesting but, in my opinion, some parts of the article could be clarified. The major strength of the study is this study practical application. Nevertheless, it requires several changes before it will be published. There are some remarks concerning this article:

1.     Five times sit to stand performance is the test, but not the movement, therefore it must be clarified its description.

2.     In my opinion, in this research it was not considered subjects BMI. Just from very first sight it could be noticeable that subjects in different groups differed by their BMI (table 1) In results part I would suggest at least to present some descriptions how this value was taken into consideration while putting subjects in some clusters.

3.     Reference list for such kind of scientific article is appropriate but some of references are too old (from last century, e.g., 5, 13, 17, 30, 33). I think it would be possible to find more recently published articles. Moreover, I found just 17 scientific articles published recently (in lats 5 years). Additionally, I found some articles cited by the same authors (e.g., 22, 26, 28, 29, 31, 37, 41).

4.     Some additional remarks:

a.      the title of article should not be presented with a dot.

b.     there are some inaccuracies in citing (line 62).

c.      Table 1 title presented below the table.

d.     kmeans function of 162 Matlab software description is also not clear - do you meant k-means clustering?

Before publication, in my opinion, article must be improved.

Round 2

Reviewer 2 Report

Minor comments:

1. I think it is okay to use linear regression to look for the relationship between UPDRS score and the a-score & f-score instead of a two-groups comparison. However, as the classification model combine both scores, can the authors provide results of the linear regression model that includes both a-score & f-score as independent variables?

2. Please check the grammar of the newly added sentences,

eg Line 206 'It was quantified the gradient obtained by the same subject ..........'

Line 341 ‘News’. Also in line 341, try to avoid using an ellipsis.
